# The prevention of adverse pregnancy outcomes by periodontal treatment during pregnancy (PROBE) intervention study—A controlled intervention study: Protocol paper

Karoline Winckler[1]*, Kathrine Hansen Bukkehave[1], Lise Tarnow[2], Peter Bindslev Iversen[2], Christian Damgaard[3], Sisse Bolm Ditlev[4], Allan Kofoed-Enevoldsen[5], Heidi Marianne Fischer[6], Signe Camilla Hjuler Dueholm[6], Jeannet Lauenborg[7], Cæcilie Trier[8], Berit Lilienthal Heitmann[1,9,10]

1 The Research Unit for Dietary Studies, The Parker Institute, Copenhagen University Hospital, Copenhagen, Denmark, 2 Steno Diabetes Center Sjælland, Herlev, Denmark, 3 Department of Odontology, Section for Oral Biology and Immunopathology, Faculty of Health and Medical Sciences, University of Copenhagen, Copenhagen, Denmark, 4 Copenhagen Center for Translational Research, Bispebjerg-Frederiksberg University Hospital, Copenhagen, Denmark, 5 Department of Endocrinology, Nykoebing Falster Hospital, Nykøbing Falster, Denmark, 6 Department of Obstetrics and Gynecology, Holbaek Hospital, Holbæk, Denmark, 7 Department of Obstetrics and Gynecology, Copenhagen University Hospital Herlev, Herlev, Denmark, 8 Department of Obstetrics and Gynecology, Nykoebing Falster Hospital, Nykøbing Falster, Denmark, 9 The Boden Group, Charles Perkins Centre, University of Sydney, Camperdown, Australia, 10 Department of Public Health, Section for General Practice, University of Copenhagen, Copenhagen, Denmark

* karoline.winckler@regionh.dk

**Data Availability Statement:** Data will be made available upon study completion and uploaded as supplementary information.

## Abstract

### Introduction

Pregnancy increases the risk of periodontitis due to the increase in progesterone and estrogen. Moreover, periodontitis during pregnancy is associated with development of pregnancy and birth related complications. The aim of this study is to determine, whether periodontal treatment during pregnancy can reduce systemic inflammation and lower the risk of adverse pregnancy and birth related outcomes.

### Methods and analysis

The PROBE study is a non-randomized controlled intervention study conducted among 600 pregnant women with periodontitis. The women will be recruited among all pregnant women at two Danish hospitals in Region Zealand during their nuchal translucency scan and will subsequently be screened for periodontitis. The intervention group includes 300 pregnant women, who will be offered state-of-the-art periodontal treatment during pregnancy. The control group includes additional 300 pregnant women, who will be offered periodontal treatment after giving birth. Outcome measures include periodontal measures, inflammatory, hormonal and glycaemic markers as well as the prevalence of preterm birth risk, low birth weight and risk markers of gestational diabetes mellitus (GDM) and preeclampsia that will

**Funding:** The study was supported by grants from the Steno Diabetes Center Sjaelland and the Sygeforsikringen Denmark Foundation (grant number 2021-0146). The funders had no role in study design, data collection and analysis, decision to publish, or preparation of the manuscript.

**Competing interests:** The authors have declared that no competing interests exist.

be collected from all screened women and further during pregnancy week 20 and pregnancy week 35 for women enrolled in the intervention.

## Ethics and dissemination

The study's findings will be published in peer reviewed journals and disseminated at national and international conferences and through social media. The PROBE study is designed to provide important new knowledge as to whether periodontal treatment during pregnancy can reduce the prevalence of complications related to pregnancy and birth.

## Clinical trials registration

The study was registered on clinicaltrials.gov (NCT06110143).

## Introduction

Periodontitis is one of the most common chronic inflammatory oral conditions affecting approximately 40–70% of the global adult population [1, 2]. Periodontitis manifests in the tooth-supporting tissues and will lead to recession of gums, tooth loosening and ultimately tooth loss if left untreated [3–5]. Accumulation of plaque (biofilm) on tooth surfaces induces gingivitis, which is still reversible although the gums can become swollen, red and may bleed. In some individuals, gingivitis progresses, and the accompanying inflammation can lead to clinical attachment loss, bone resorption and loosening of teeth, which is irreversible and also marks the onset of periodontitis. If untreated, periodontitis may also result in several complications including impaired masticatory function, reduced self-esteem and reduced interest in social interactions [6]. Accumulating evidence has linked periodontitis to a wide range of common medical conditions such as diabetes, cardiovascular disease, rheumatoid arthritis and Alzheimer's disease [7].

Pregnant women are susceptible to develop periodontitis due to hormonal and immunological changes related to the pregnancy status [8]. The immunologic changes have recently been revealed as mediators of a pathogenic shift in the oral microbiome emerging during pregnancy [9]. Studies have linked low-grade inflammation to periodontitis, also during pregnancy, by an increased vascular permeability, skewed immune responses and a shift in the composition of the periodontal microbiota [10]. Consequently, maternal periodontitis may influence offspring health through several mechanisms, including the exposure to increased inflammatory load, and creating of a dysbiosis of intrauterine microenvironment and epigenomic changes in the offspring [11, 12]. Emerging evidence, including results from a systematic review, has shown that periodontitis during pregnancy can increase the risk of premature birth (< 37 weeks gestation) and low birthweight (< 2.5 kg) [13, 14]. Studies suggest that treatment during pregnancy might reduce the risk of preterm birth, low birthweight and potentially reduce the risk of both pre-eclampsia and gestational disease [15, 16] However, studies are still debating the exact effects [17], and the biological mechanisms linking periodontitis to adverse pregnancy outcomes are not yet fully understood [18].

The prevalence of periodontitis during pregnancy has not been systematically assessed in many countries, including Denmark, and the prevalence of periodontitis from the few existing studies vary significantly with a range from 12% [19, 20] to 61% [21]. The large variation might, partly, be attributed to diverse definitions of periodontitis [22]. Still, optimal oral health

care during pregnancy is often neglected, and data from the Danish Dental Association indicate that as many as 40% of Danish women in childbearing age do not visit a dentist regularly [20].

Treatment of periodontitis in pregnant women has previously been found to be safe and to have beneficial effects beyond the oral cavity for both the mother and the child [17]. Reducing complications, like gestational diabetes and preeclampsia during pregnancy, and fetal growth restriction are expected to beneficially influence the risk of later development of metabolic disturbances in the offspring [23–26]. Another advantage of treating periodontitis during pregnancy is the potential long-term reduction in risk of other chronic diseases such as diabetes, cardiovascular disease and Alzheimer's [27].

If the findings from the PROBE study show an effect of periodontal treatment on the risk of pregnancy and birth related outcomes, the main message will be that oral hygiene, regular visits to a dentist and treatment of existing periodontitis during pregnancy should be given higher priority. Some countries, like the UK and Brazil, acknowledging the importance of good oral hygiene during pregnancy, currently offer free dental care during pregnancy. In Denmark dental care and treatment is public financed and currently free of charge for children up to the age of 21 years. Hereafter, only partly (around 40% reimbursement) subsidised. A recent study [28] on the financial consequences of providing free dental care to pregnant women suggests that the long-term beneficial effects both for the women and the offspring may be significant and justify the costs of providing treatment without patient out of pocket costs. The PROBE study plans to assess this aspect through an economic analysis. This analysis will quantify the cost-effectiveness of the preventive intervention in the PROBE study using analyses suggested by Cookson et al. 2017 [29].

## The overall rational and aim of the PROBE intervention study is

To determine the effects of the periodontal treatment intervention on a) the risk of preterm birth, low birthweight and markers of preeclampsia and gestational diabetes; and on b) changes in markers of systemic inflammation related to periodontal treatment during and after pregnancy.

## Materials and methods

### Design

Approval was granted by the Committees on Health Research Ethics in the Capital Region of Denmark (journal number H-20083249) on May 19, 2021 and by the Danish Data Protection Agency. The study will be conducted in accordance with the Helsinki Declaration and guidelines for Good Clinical Practice. The study was registered on clinicaltrials.gov (NCT06110143). The PROBE study is a controlled intervention study, where approximately 1200 pregnant women are recruited from Holbaek Hospital and Nykoebing Falster Hospital in Region Zealand in Denmark. About half of the participants are expected to have periodontitis, based on findings from other Nordic countries (1). The recruitment will take place in relation to their appointment for the nuchal translucency scan in gestational week 11–14 followed by a periodontal screening visit with a licensed dentist. The secretaries orally inform all pregnant women attending the nuchal translucency scan about the study. The pregnant women who participate will give informed written consent. Inclusion criteria for the intervention are pregnant women, who speak and understand Danish and are scheduled to give birth and either Holbaek Hospital or Nykoebing Falster Hospital, and are diagnosed with periodontitis (stage I-III) by the study dentist. Exclusion criteria are pregnant women without periodontitis,

pregnant women with less than 20 healthy teeth, pregnant women with periodontitis who receives periodontal treatment and pregnant women with periodontitis stage IV.

Data from the Danish Dental Association (2019) indicate that approximately 35% of Danish women (18–39 years) with regular dentist visits have periodontitis. In the same age group, an estimated 65% of the women living in Region Zealand have regular dentist visits. There is no information about periodontitis for those 35%, who do not visit a dentist regularly. Recruitment of pregnant women with periodontitis into the study will last over a period of 18 months for the intervention group and an additional 18 months for the control group. Over a period of 18 month, we expect that approximately 600 pregnant women in the region are eligible for inclusion. Provided half of the invited women accepts the invitation to participate we can include 300 women in each of the two study arms.

The intervention group receives state-of-the-art periodontal treatment during pregnancy, and the control group receives periodontal treatment after giving birth. The method and data collection activities are the same for both the intervention and the control group, except for the timing of the periodontal treatments that are performed during pregnancy for the intervention group and after giving birth for the control group (Fig 1). The periodontal treatment will be up to approximately 20 weeks for both groups: Each included woman can receive up to five periodontal treatments.

Demographic information, data related to mental health, use of medicine, diet intake, smoking status, physical activity, socioeconomic status, oral health care and dental habits will be collected at baseline (gestational week 11–13) and at gestational week 35 via online questionnaires. Two months and six months post-partum additional questionnaires addressing the birth information, growth of the baby, breastfeeding, educational and employment status of the partner, weight and height of the partner, current weight of the mother, oral health and dental habits of the mother, and level of contentment for participating in the study will be sent online to all enrolled participants (S1 Appendix).

Dental treatment takes place at regional dental clinics in Holbaek, Kalundborg, Vordingborg and Maribo, all in Region Zealand, close to the participants residence, in order to facilitate a potential implementation of the project afterwards in the antenatal care program.

## Patient and public involvement

A small pilot study was initially conducted in the planning phase of the study and included vulnerable pregnant women. The included pregnant women, who did not see a dentist regularly (defined as no dental visits within the last 18 months and/or 2 or less dental visits within the last 5 years) informed that they would be willing to participate in a study like the PROBE study and believed that oral health should be given more attention during pregnancy and be implemented in the antenal care program. Another finding was that the two primary barriers for why these pregnant women in region Zealand do not see a dentist on a regular basis were economic constraints and a general lack of priority of going for regular dental visits.

Offering free dental care during pregnancy in Denmark may be an easy implementable strategy to reduce the burden of adverse birth outcomes. There will be economic expenses related to providing free dental care, but the long-term beneficial effects both for the women and children are potentially significant, and a cost effectiveness analysis is scheduled to be performed based on our results to further support this. After the PROBE intervention study has been completed a fourth and final questionnaire will be sent to all the included participants with questions addressing the participants perspectives and opinions on participating in the study.

| | STUDY PERIOD | | | | | | | | | | | | |
|---|---|---|---|---|---|---|---|---|---|---|---|---|---|
| | Enrollment | Screening | Periodontal treatments | | | | | | | | | | |
| **TIMEPOINT*** | *11-13** | *13-20** | *15-36** | | | | | *Birth* | *0-24 post-partum* | | | | |
| **ENROLLMENT:** | | | | | | | | | | | | | |
| **Eligibility screen** | X | | | | | | | | | | | | |
| **Informed consent** | X | | | | | | | | | | | | |
| *Dental screening* | | X | | | | | | | | | | | |
| **Allocation** | X | | | | | | | | | | | | |
| **INTERVENTIONS:** | | | | | | | | | | | | | |
| *Periodontal treatment during pregnancy* | | | X | X | X | X | X | | | | | | |
| *Postpartum periodontal treatment* | | X | | | | | | X | X | X | X | X | X |
| **ASSESSMENTS:** | | | | | | | | | | | | | |
| *Demographic baseline variables* | X | X | | | | | | | | | | | |
| *Pregnancy and birth related outcomes* | | | | | | | | X | | | | | |
| *Blood samples* | X | | X | | | | X | | | | | | |
| *Questionnaires* | X | | | | | | X | | | X | | | X |
| *Saliva* | X | | | | | | X | | | | | | |

*Gestation week

**Fig 1. SPIRIT schedule of enrollment, screening, and interventions.**

## Outcome measures

Primary outcome measures include gestational length (days) and birth weight (g). Preterm birth is defined as birth before gestational week 37 and rates will be described below gestational week 37 and 34. Secondary outcomes include risk markers for preeclampsia (defined as hypertension and/or proteinuria) and gestational diabetes (defined as glucose intolerance).

Long term outcome measures include growth development, infections, use of medicine and hospitalizations. These measures will be collected and retrieved through the general practitioner and from medical journals until the age of 2 years.

## Periodontal measures

The same study dentist will be performing all the periodontal examinations. Periodontal clinical parameters used to make the periodontal diagnosis are 'plaque index' (PI), 'bleeding on probing (BOP)', 'periodontal pocket depth (PPD)' and 'clinical attachment loss (CAL)' and

will be measured at baseline (gestational week 11–20), and at gestational week 35–36 in the intervention group. PI will be measured during treatment (up to four treatment sessions before gestational week 35).

PI will be calculated as the mean of the amount of dental plaque present on each tooth. BOP will be calculated as the mean gingival bleeding observed 10 seconds after probing and will be recorded as a percentage. CAL will be measured (in 'mm') as the distance from the cemento-enamel junction to the base of the gingival sulcus [30]. PPD will be measured (in mm) as the distance from the gingival margin to the base of the gingival sulcus [30].

As per the clinical definition of periodontitis, a patient will be diagnosed with periodontitis, if a CAL of at least 1 mm is observed on at least two non-adjacent teeth, or a facial/ lingual CAL of at least 3mm is observed on at least two teeth (which may be adjacent) [31]. The observed CAL must not be due to reasons other than periodontitis e.g., traumatic gingival recessions, cervical dental caries, endodontic lesion draining through the marginal periodontium, vertical root fracture or complications associated with the second molar [31]. Periodontitis will further be classified into stages as per the severity of the disease, defined by the worst affected tooth with the highest observed CAL. Due to the lack of radiological diagnosis, CAL, PPD and number of missing teeth, will be the diagnostic criteria for staging periodontitis: StageI(CAL = 1–2 mm, PPD $\leq$ 4 mm, no missing teeth), StageII (CAL = 3–4 mm, PPD $\leq$ 5 mm, no missing teeth), Stage III (CAL $\geq$ 5 mm, PPD $\geq$ 6 mm, 1–4 missing teeth) and Stage IV (CAL $\geq$ 5 mm, PPD $\geq$ 6 mm, $\geq$ 5 missing teeth) [31].

## Blood samples

The recruited women will have blood samples collected at either Holbaek or Nykoebing Falster Hospital at the Department of Clinical Biochemistry. Centrifuged and isolated plasma will be stored in Region Zealand Biobank for later analyses. Analyses of the blood samples will be done at Copenhagen Center for Translational Research at the Capital Region of Denmark. Blood will be taken at three time points, at baseline in gestational week 11–13 (from all recruited participants, n = 1200) and in gestational week 20–22 and at gestational week 35–37 from the included women (n = 600).

Inflammatory markers interleukin (IL)- 1β, IL-6, IL-10, IL-17α, (hs-CRP) and tumour necrosis factor (TNF)-α, hormones (insulin, leptin, growth differentiation factor (GDF)-15, adiponectin, resistin) and glycemic markers haemoglobin A1c (HbA1c) will be measured, and the blood samples will be analyzed in one batch at the end of the study to minimize variation. CRP and HbA1c will be measured and analysed on the same day the samples are taken. Saliva samples will be collected at baseline in gestational week 11–13 from all recruited participants and again at gestational week 35–37 from the included women. Procedures are the same for all women in both the intervention and the control group.

## Analyses of blood samples

Blood will be drawn by venipuncture into 2 ml EDTA tubes. The blood will be mixed by inversion and centrifuged at 900g in 15 min. The supernatant will be isolated, aliquoted, and stored at −80˚C until analysis. Plasma levels of the proinflammatory cytokine IL-1β, IL-6, IL-10, IL-17α, TNF-α will be measured in duplicate by operators and blinded to the clinical state of the participant using Meso Scale Discovery (MSD; Rockville, MD) according to the manufacturer's instruction. S-Plex, cat# 151B3S-1 (IL-6), cat# 151C3S-1 (IL-17α), cat# K151E3S-1 (TNF-α) and cat# K151Y2S-1 (IL-10). Analysis will be conducted on the MSD QuickPlex SQ 120 with software and the biomarker absolute values reported as absolute concentrations in picogram per milliliter (pg/ml).

**Table 1. Power calculation.**

| Outcomes | Expected mean (SD) | Minimal detectable effect (n = 600) | Minimal detectable effect (n = 400) |
|---|---|---|---|
| Gestational length (days) | 272 (9) | 2.1 | 2.5 |
| Birth Weight (g) | 3490 (567) | 130 | 159 |

## Power calculation

Statistical power was evaluated prior to commencement of the study using a two-sample means test. Minimal detectible effects for key study outcomes; gestational length (days) and birth weight (g) at 80% power and a significance level at 0.05, assuming 600 (number of enrolled) or 400 participants (200 participants with dropout and/or missing information), are presented below (Table 1).

## Statistics

Baseline characteristics of participants will be summarized using descriptive statistics (i.e. means and standard deviations for normally distributed continuous variables, medians and ranges for skewed continuous variables, frequencies and proportions for categorical/binary variables), for both treatment arms and for the total sample. All analyses will be conducted according to the intention-to-treat approach, including all participants, regardless of adherence to the study protocol. The effect of the periodontal treatment performed during pregnancy will be analyzed with regression analyses adjusted for relevant confounders (age, BMI, parity, smoking status, medicine use, socioeconomic status and education level). In addition, statistical analyses (multiple regression analysis, potentially spline analysis) of the changes in hormonal and inflammatory markers in both treatment arms will be performed. To determine the effects of the intervention, time and group-time interaction on outcomes (inflammatory and hormonal markers, glycemic control and periodontitis status), one-way repeated measures ANOVA will be applied. Procedure for testing an interaction effect: If the interaction effect is significant, provide an interpretation of the results, but do not test main effects because the tests for main effects are uninteresting in light of significant interactions. If interaction effects are non-significant, drop the interaction effects from the model and test the main effects. Determining which results to present when testing interactions is often a multi-step process. Statistical software (SPSS, (version 19, Chicago IBM) will be used. A P-value $< 0.05$ will be considered statistically significant.

All data will be stored in password-controlled databases, and all data will be encrypted. All data collected will be anonymized using a unique study ID. Any identifiable data (e.g. consent forms) will be stored in secure, locked facilities with access limited to the study team.

## Discussion

To ensure dissemination, the findings of the study will be published in peer reviewed journals and disseminated at national and international conferences. Furthermore, the results will be communicated via social media, presented at academic meetings, and shared with participating communities as well as with national and international policymaking bodies. The REDCap platform, which will be used to collect demographic data, measures from the periodontal examinations, and data from the qualitative questionnaires, is a secure, web-based application designed to enable responses to be automatically uploaded into safe data storage to maximize data security. The safe data storage has been certified and conforms to the Danish National Health Service Information Governance Toolkit.

The decision to participate in the study is voluntary and will not affect the patient's clinical care. The patients can withdraw consent at any given point in time, also after data have been collected. Patients will be provided with e-mail details to the research team, providing the opportunity to withdraw without having to speak to anyone involved in the PROBE study.

We envisage that the outputs of the study will include:

- A deeper understanding of the effects of periodontal treatment on pregnancy and birth related complications in pregnant Danish women with periodontitis.

- Knowledge about the mediating inflammation markers.

- Relatively long-term follow up effects on health parameters of the offspring (growth development, infections, use of medicine and hospitalizations).

- An overview of the prevalence of periodontitis in pregnant women.

## Supporting information

**S1 Checklist. SPIRIT 2013 checklist: Recommended items to address in a clinical trial protocol and related documents\*.**
(DOC)

**S1 Appendix. Data retrieved from questionnaires.**
(DOCX)

## Acknowledgments

The authors would like to thank the participants in this study, the vulnerable pregnant women, who took the time during this period in their life to participate in this study. We would also like to acknowledge the staff at Department of Obstetrics and Gynecology at Holbaek and Nykoebing Falster Hospitals in Region Zealand, Denmark, for their contributions to the data collection process.

## Author Contributions

**Conceptualization:** Karoline Winckler, Lise Tarnow, Peter Bindslev Iversen, Christian Damgaard, Allan Kofoed-Enevoldsen, Heidi Marianne Fischer, Signe Camilla Hjuler Dueholm, Jeannet Lauenborg, Cæcilie Trier, Berit Lilienthal Heitmann.

**Formal analysis:** Karoline Winckler, Berit Lilienthal Heitmann.

**Funding acquisition:** Karoline Winckler.

**Investigation:** Karoline Winckler, Kathrine Hansen Bukkehave, Lise Tarnow, Christian Damgaard, Sisse Bolm Ditlev, Allan Kofoed-Enevoldsen, Heidi Marianne Fischer, Signe Camilla Hjuler Dueholm, Jeannet Lauenborg, Cæcilie Trier.

**Methodology:** Karoline Winckler, Kathrine Hansen Bukkehave, Peter Bindslev Iversen, Sisse Bolm Ditlev, Berit Lilienthal Heitmann.

**Project administration:** Karoline Winckler.

**Software:** Berit Lilienthal Heitmann.

**Validation:** Karoline Winckler, Kathrine Hansen Bukkehave, Peter Bindslev Iversen, Sisse Bolm Ditlev, Berit Lilienthal Heitmann.

**Writing – original draft:** Karoline Winckler.

**Writing – review & editing:** Kathrine Hansen Bukkehave, Lise Tarnow, Peter Bindslev Iversen, Christian Damgaard, Sisse Bolm Ditlev, Allan Kofoed-Enevoldsen, Heidi Marianne Fischer, Signe Camilla Hjuler Dueholm, Jeannet Lauenborg, Cæcilie Trier, Berit Lilienthal Heitmann.

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
