## [Decision Letter · Decision Letter 0]

16 Feb 2024

PONE-D-23-40210The prevention of adverse pregnancy outcomes by periodontal treatment during pregnancy (PROBE) intervention study - A controlled intervention study: protocol paperPLOS ONE

Dear Dr. Winckler,

Thank you for submitting your manuscript to PLOS ONE. After careful consideration, we feel that it has merit but does not fully meet PLOS ONE’s publication criteria as it currently stands. Therefore, we invite you to submit a revised version of the manuscript that addresses the points raised during the review process.

We look forward to receiving your revised manuscript.

Kind regards,

Gaetano Isola, Ph.D.

Academic Editor

PLOS ONE

Journal Requirements:

Reviewers' comments:

Reviewer's Responses to Questions

**Comments to the Author**

1. Does the manuscript provide a valid rationale for the proposed study, with clearly identified and justified research questions?

Reviewer #1: Yes

Reviewer #2: Yes

Reviewer #3: Yes

2. Is the protocol technically sound and planned in a manner that will lead to a meaningful outcome and allow testing the stated hypotheses?

Reviewer #1: Yes

Reviewer #2: Yes

Reviewer #3: Partly

3. Is the methodology feasible and described in sufficient detail to allow the work to be replicable?

Reviewer #1: Yes

Reviewer #2: Yes

Reviewer #3: No

4. Have the authors described where all data underlying the findings will be made available when the study is complete?

Reviewer #1: No

Reviewer #2: No

Reviewer #3: Yes

5. Is the manuscript presented in an intelligible fashion and written in standard English?

Reviewer #1: Yes

Reviewer #2: Yes

Reviewer #3: Yes

6. Review Comments to the Author

You may also provide optional suggestions and comments to authors that they might find helpful in planning their study.

Reviewer #1: Thank you for presenting your protocol.

I have only one suggestion: Primary outcomes should include the rates of preterm birth (below 37 weeks and below 34 weeks), because this is a more clinically useful parameter, and should improve your work acceptance.

Reviewer #2: A two-arm controlled study is being proposed where the aim is to determine whether periodontal treatment during pregnancy can reduce systemic inflammation and lower risk of complications during pregnancy.

Minor revisions:

1- Abstract: “The aim of this study is to determine whether periodontal treatment during pregnancy can reduce systemic inflammation and lower risk of complications during pregnancy and fetal growth.” The “and fetal growth” phrase in this sentence is confusing.

2- In the abstract, clarify if the study will be randomized or not.

3- The standard statistical term for average is mean.

4- Line 226: Power calculation. Indicate the statistical testing method used to estimate the power in the scenarios.

5- Line 240: List the confounders.

6- Line 240: Describe the statistical methods that will be applied to analyze changes in hormonal and inflammatory markers.

7- Procedure for testing an interaction effect: If the interaction effect is significant, provide an interpretation of the results, but do not test main effects because the tests for main effects are uninteresting in light of significant interactions. If interaction effects are non-significant, drop the interaction effects from the model and test the main effects. Determining which results to present when testing interactions is often a multi-step process.

8- Identify the software that will be used for statistical analysis.

Reviewer #3: In the manuscript entitled: " The prevention of adverse pregnancy outcomes by periodontal treatment during pregnancy (PROBE) intervention study - A controlled intervention study: protocol paper" the authors aimed to assess whether periodontal treatment during pregnancy can reduce systemic inflammation and lower risk of complications during pregnancy and fetal growth.

The authors concluded that Furthermore, the results will be communicated via social media, presented at academic meetings, and shared with participating communities as well as with national and international policymaking bodies. The REDCap platform, which will be used to collect demographic data, measures from the periodontal examinations, and data from the qualitative questionnaires, is a secure, web-based application designed to enable responses to be automatically uploaded into safe data storage to maximize data security.

Major comments:

In general, the idea and innovation of this study regards the analysis of oral and periodontal health in pregnancy is interesting and novel because the role these aspects in medicine are validated but further studies on this topic could be an innovative issue in this field could be open a creative matter of debate in literature by adding new information. Moreover, there are few reports in the literature that studied this interesting topic with this kind of study design.

The study was well conducted by the authors; However, there are some concerns to revise that are described below.

The introduction section resumes the existing knowledge regarding the important factor linked with the impact mediators involved together oral health and with periodontitis.

However, as the importance of the topic, the reviewer strongly recommends, before a further re-evaluation of the manuscript, to update the literature through read, discuss and must cites in the references with great attention all of those recent interesting articles, that helps the authors to better introduce and discuss the role of mediators (NT-PRO-BNP and TGF beta 1) in periodontitis and related recessions by adding as a references these article, before any further assessment of the manuscript: 1) DOI: doi: 10.1002/JPER.23-0063. PMID: 37433155; 2) DOI: 10.1177/1721727X1301100217

The authors should be better specified, at the end of the introduction section, the rationale of the study and the aim of the study. In the central section, should better clarify inclusions and exclusions criteria of the selected sample.

Please better state the results obtained in the abstract.

The discussion section appears well organized with the relevant paper that support the conclusions, even if the authors should better discuss the relationship regarding the by periodontitis in and risk of oxidative stress evolution that could improve the quality of life in periodontitis patients which undergo pregnancy. The conclusion should reinforce in light of the discussions.

In conclusion, I am sure that the authors are fine clinicians who achieve very nice results with their adopted protocol. However, this study, in my view does not in its current form satisfy a very high scientific requirement for publication in this journal and requests a revision before a futher re-evaluation of the manuscript.

Minor Comments:

Abstract:

-Better formulate the abstract section by better describing the aim of the study

Introduction:

-Please refer to major comments

Discussion

-Please add a specific sentence that clarifies the results obtained in the first part of the discussion

7. PLOS authors have the option to publish the peer review history of their article (what does this mean?). If published, this will include your full peer review and any attached files.

Reviewer #1: No

Reviewer #2: No

Reviewer #3: No

---

## [Author Response · Author response to Decision Letter 0]

13 Mar 2024

Please find responses to reviewer and editor comments attached as a seperate file.

---

## [Decision Letter · Decision Letter 1]

26 Mar 2024

The prevention of adverse pregnancy outcomes by periodontal treatment during pregnancy (PROBE) intervention study - A controlled intervention study: protocol paper

PONE-D-23-40210R1

Dear Dr. Winckler,

We’re pleased to inform you that your manuscript has been judged scientifically suitable for publication and will be formally accepted for publication once it meets all outstanding technical requirements.

Kind regards,

Gaetano Isola, Ph.D.

Academic Editor

PLOS ONE

Additional Editor Comments (optional):

The authors have well addressed all issuea raised by the reviewers

Reviewers' comments:

Reviewer's Responses to Questions

**Comments to the Author**

1. Does the manuscript provide a valid rationale for the proposed study, with clearly identified and justified research questions?

Reviewer #2: Yes

Reviewer #3: Yes

2. Is the protocol technically sound and planned in a manner that will lead to a meaningful outcome and allow testing the stated hypotheses?

Reviewer #2: Yes

Reviewer #3: Yes

3. Is the methodology feasible and described in sufficient detail to allow the work to be replicable?

Reviewer #2: Yes

Reviewer #3: Yes

4. Have the authors described where all data underlying the findings will be made available when the study is complete?

Reviewer #2: Yes

Reviewer #3: Yes

5. Is the manuscript presented in an intelligible fashion and written in standard English?

Reviewer #2: Yes

Reviewer #3: Yes

6. Review Comments to the Author

You may also provide optional suggestions and comments to authors that they might find helpful in planning their study.

Reviewer #2: All comments have been adequately addressed.

Reviewer #3: In this revised version of the manuscript, the authors have well addressed all issues raised by the reviewer. The manuscript can be accepted for publication.

7. PLOS authors have the option to publish the peer review history of their article (what does this mean?). If published, this will include your full peer review and any attached files.

Reviewer #2: No

Reviewer #3: No
